# PDE6D Mediates Trafficking of Prenylated Proteins NIM1K and UBL3 to Primary Cilia

**DOI:** 10.3390/cells12020312

**Published:** 2023-01-13

**Authors:** Siebren Faber, Stef J. F. Letteboer, Katrin Junger, Rossano Butcher, Trinadh V. Satish Tammana, Sylvia E. C. van Beersum, Marius Ueffing, Rob W. J. Collin, Qin Liu, Karsten Boldt, Ronald Roepman

**Affiliations:** 1Department of Human Genetics, Radboud Institute for Molecular Life Sciences, Radboud University Medical Center, 6525 GA Nijmegen, The Netherlands; 2Division of Experimental Ophthalmology and Medical Proteome Center, Center of Ophthalmology, University of Tübingen, 72076 Tübingen, Germany; 3Department of Ophthalmology, Ocular Genomics Institute, Massachusetts Eye and Ear, Harvard Medical School, Boston, MA 02115, USA; 4Department of Human Genetics, Donders Institute for Brain, Cognition and Behaviour, Radboud University Medical Center, 6525 GA Nijmegen, The Netherlands

**Keywords:** PDE6D, FAM219A, NIM1K, UBL3, protein trafficking, prenylation, cilium, photoreceptor

## Abstract

Mutations in PDE6D impair the function of its cognate protein, phosphodiesterase 6D (PDE6D), in prenylated protein trafficking towards the ciliary membrane, causing the human ciliopathy Joubert Syndrome (JBTS22) and retinal degeneration in mice. In this study, we purified the prenylated cargo of PDE6D by affinity proteomics to gain insight into PDE6D-associated disease mechanisms. By this approach, we have identified a specific set of PDE6D-interacting proteins that are involved in photoreceptor integrity, GTPase activity, nuclear import, or ubiquitination. Among these interacting proteins, we identified novel ciliary cargo proteins of PDE6D, including FAM219A, serine/threonine-protein kinase NIM1 (NIM1K), and ubiquitin-like protein 3 (UBL3). We show that NIM1K and UBL3 localize inside the cilium in a prenylation-dependent manner. Furthermore, UBL3 also localizes in vesicle-like structures around the base of the cilium. Through affinity proteomics of UBL3, we confirmed its strong interaction with PDE6D and its association with proteins that regulate small extracellular vesicles (sEVs) and ciliogenesis. Moreover, we show that UBL3 localizes in specific photoreceptor cilium compartments in a prenylation-dependent manner. Therefore, we propose that UBL3 may play a role in the sorting of proteins towards the photoreceptor outer segment, further explaining the development of PDE6D-associated retinal degeneration.

## 1. Introduction

The primary cilium is an antenna-like organelle that projects from the plasma membrane into the extracellular space, functioning as a sensory transducer in nearly all vertebrate cell types [1]. The primary cilium originates from the ciliary basal body, the former mother centriole, that docks at the cell membrane as the ciliary basal body upon cellular quiescence, and there functions as the nucleation site for the assembly of the ciliary microtubule-based axoneme [2]. The protruding axoneme is encompassed by a ciliary membrane, which is contiguous with the plasma membrane [3]. The proteins active in cilia have to be imported via transport across a selective barrier called the transition zone [4].

One of the key proteins regulating protein transport towards the ciliary membrane is phosphodiesterase 6D (PDE6D). More specifically, PDE6D is involved in the transport of prenylated proteins [5]. Protein prenylation or lipidation is a post-translational protein modification that comprises the transfer of either a farnesyl or a geranylgeranyl moiety to a C-terminal cysteine of a specific target protein [6,7]. Through binding to these lipid groups, PDE6D can solubilize the prenylated protein, thereby facilitating its transport through the hydrophilic cytosol towards the ciliary membrane [8]. At the destination membrane, the small GTPases ARL2 and ARL3, in their GTP-bound state, interact with PDE6D [9]. Consequently, the lipid group of the cargo protein is squeezed out due to a size reduction of the hydrophobic cavity of PDE6D, followed by membrane integration of the prenylated protein [10].

The importance of the transport function of PDE6D becomes apparent when the gene encoding PDE6D is mutated. Mutations in *PDE6D* are associated with Joubert syndrome (JS), a severe syndromic neuronal ciliopathy resulting in developmental cerebellar malformations, classically diagnosed by the ‘molar tooth sign’ upon MRI [11,12]. A common additional feature of *PDE6D*-associated JS is retinal degeneration [13]. The important role of PDE6D in photoreceptors is demonstrated by *Pde6d* knockout mice. Apart from a reduced body weight, *Pde6d^−/−^* mice solely develop recessive rod-cone dystrophy [14]. The rod-cone dystrophy was shown to be caused by impaired transport of prenylated proteins from the photoreceptor inner segment towards the photoreceptor outer segment, indicating its conserved function among various cell types. In addition, the genes encoding several prenylated proteins that are trafficked by PDE6D, including RPGR and RAB28, are also mutated in patients with non-syndromic inherited retinal degeneration [15,16,17].

Despite the known function of PDE6D in prenylated cargo trafficking towards the membrane of cilia, including highly specialized photoreceptor cilia, a comprehensive, in-depth overview of the conserved prenylated cargo of PDE6D is still lacking. The generation of such an overview will give more insights into the PDE6D-associated disease mechanisms, including inherited blindness.

Therefore, we set out to study the PDE6D proteome in two different cell lines from two different species by implementing tandem affinity proteomics. Among the most significantly enriched proteins in both datasets were prenylated proteins involved in GTPase activity, nuclear protein import, ubiquitination, and photoreceptor integrity. The generated datasets allowed us to identify novel ciliary cargo proteins of PDE6D, including FAM219A, serine/threonine-protein kinase NIM1 (NIM1K), and ubiquitin-like protein 3 (UBL3). We show that NIM1K and UBL3 localize inside the cilium in a prenylation-dependent manner. Furthermore, UBL3 localizes in vesicle-like structures around the base of the cilium. Through affinity proteomics of UBL3, we confirmed its strong interaction with PDE6D and identified an interactome with strong links to small extracellular vesicles (sEVs) and ciliogenesis. Moreover, we show that UBL3 localizes in specific photoreceptor cilium compartments in a prenylation-dependent manner. Therefore, we propose that UBL3 may play a role in ciliary small extracellular vesicle (sEV) regulation, an important process in photoreceptor outer segment formation [18].

## 2. Materials and Methods

### 2.1. Cell Culture

The human embryonic kidney 293T (HEK293T) cells were cultured in Dulbecco’s modified Eagle’s medium (DMEM, Sigma-Aldrich, Amsterdam, The Netherlands, D0819) with 1% Penicillin-Streptomycin solution (Sigma-Aldrich, P4333), 1% sodium pyruvate (Sigma-Aldrich, S8636), and 10% fetal bovine serum (FBS, Sigma-Aldrich, F7524). The murine inner medullary collecting duct-3 (mIMCD3) cells were cultured in DMEM (Sigma-Aldrich, D0819)/F12 (Sigma-Aldrich, N6658) in a 1:1 ratio containing the same supplements. The mIMCD3 Flp-in cells were cultured in DMEM (Sigma-Aldrich, D0819)/F12 (Sigma-Aldrich, N6658) in a 1:1 ratio containing the same supplements with the addition of Zeocin (Thermo Fisher, Breda, The Netherlands, 100 µg/mL). The cultures were incubated at 37 °C with 5% CO_2_. Further, the cell lines were tested for mycoplasma contamination (Minerva Biolabs, Berlin, Germany, Venor GeM Mycoplasma Detection Kit, 11-1050) prior to use.

### 2.2. Plasmid Construction

The human full-length *PDE6D* (O43924-1), *FAM219A* wild-type (WT; Q8IW50-1), *NIM1K* WT (Q8IY84), and *RAF1* (P04049-1) complementary DNA (cDNA) clones were amplified from human fetal brain marathon-ready cDNA (Takara, Paris, France, 639302). The *FAM219A* C > A (C182A), *NIM1K* C > A (C433A), and *NIM1K* S > K (S430K) cDNA clones were amplified using the WT vectors and specific primers containing the desired mutation. The *UBL3* WT (Q9Z2M6-1) and *UBL3* C > A (C113/114A) cDNA clones were kindly donated by Prof. Kunihiro Tsuchida, Japan [19]. In addition, *UBL3* S > K (S111K) was amplified using the WT vector and specific primers containing the desired mutation. The generated entry clones were cloned into various destination vectors by Gateway (GW) cloning (Thermo Fisher). The destination vectors that were used to generate expression vectors encoding PDE6D, FAM219A WT/C > A, NIM1K WT/C > A/S > K, UBL3 WT/C > A/S > K, and RAF1 (control) are shown in Table 1.

### 2.3. Generation of Stable Cell Lines

The stable cell lines were created using the mIMCD3 Flp-In system according to the manufacturer’s protocol (Thermo Fisher). In short, mIMCD3 Flp-In cells were transfected with expression vectors encoding PDE6D, UBL3 WT, UBL3 C > A, or RAF1 (control) and N-terminally tagged with N-TAP or eGFP. After 40 h, cultures were selected in DMEM/F12 containing Hygromycin (400 μg/mL). The expanded cell lines were assessed by affinity proteomics and immunofluorescence microscopy.

### 2.4. Affinity Proteomics

The HEK293T cells were transfected with expression vectors encoding N-TAP-tagged PDE6D or RAF1 (control). The protein RAF1 was selected as the control since it has been shown to be a valid non-ciliary control of relatively average protein size (648aa) with modest recombinant expression levels [24]. Furthermore, RAF1 also functions as a positive/quality control, since we exactly know, based on previous experiments [24], which proteins should be detected after mass spectrometry analysis, in contrast to an empty vector.

After 48 h, cells were lysed in lysis buffer containing 0.5% Nonidet-P40, a protease inhibitor cocktail (Roche, Basel, Switzerland), and phosphatase inhibitor cocktails II and III (Sigma-Aldrich) in TBS (30 mM Tris-HCl, pH 7.4, and 150 mM NaCl) for 30 min at 4 °C, and cleared by centrifugation at 10,000× *g* for 15 min. The cleared lysates were transferred to Strep-Tactin Superflow resin (IBA-Lifesciences, Göttingen, Germany) and incubated for 1.5 h at 4 °C. Subsequently, the affinity gel with bound protein complexes was washed three times with wash buffer (TBS containing 0.1% NP40 and phosphatase inhibitor cocktails II and III), followed by elution with Strep-Tactin elution buffer with desthiobiotin (IBA-Lifesciences). The eluates were transferred to an anti-FlagM2 affinity gel (Sigma-Aldrich) and incubated for 1.5 h at 4 °C. Subsequently, the affinity gel with bound protein complexes was washed three times with wash buffer (TBS containing 0.1% NP40 and phosphatase inhibitor cocktails I and II), followed by two times with 1× TBS. The protein complexes were eluted with Flag peptide (200 μg/mL; Sigma-Aldrich) in TBS. After purification, the samples were precipitated with methanol and chloroform, as previously described [24]. The precipitated protein pellets were stored at −80 °C until mass spectrometry (MS) analysis.

The mIMCD3 Flp-in cells stably expressing PDE6D, RAF1, UBL3 WT, or UBL3 C > A tagged with N-TAP were subjected to serum deprivation (culturing medium with 0.2% FBS) for 48 h to induce ciliogenesis. A tandem affinity approach was performed, as described above, for the cells stably expressing PDE6D or RAF1 (control). Further, a one-step FLAG purification was performed for the cells stably expressing UBL3 WT or UBL3 C > A, with cells stably expressing RAF1 used as a control.

### 2.5. MS Analysis, Protein Quantification and Statistics

The precipitated proteins were subjected to in-solution tryptic cleavage and StageTip purification (Thermo Fisher), followed by MS analysis on an Orbitrap Fusion Tribrid mass spectrometer (Thermo Fisher), as described earlier [25]. The label-free quantification (LFQ) was performed using Maxquant (v.1.6.1.0), and identified proteins were analyzed using Perseus (v.1.6.2.3) [26,27]. All data were filtered for potential contaminants, peptides only identified by site, and reverse database identifications. The proteins were filtered to be present in ≥3 of the 4 or 5 replicates. Further, for PDE6D versus RAF1 (control) and vice versa, a one-sided two-sample test with permutation-based FDR correction was performed (FDR ≤ 0.05). In case of, UBL3 WT versus RAF1 (control) and UBL3 C113/114A, a one-sided two-sample test with (Tier I) and without (Tier II) permutation-based FDR correction was performed (FDR ≤ 0.05; *p* ≤ 0.05). Proteins were considered significantly enriched when they passed the two-sample test and showed an enrichment of at least 32-fold (log2 = 5).

In order to find out which of the significantly enriched proteins are prenylated, we applied the following criteria: Proteins containing the Ca_1_a_2_X sequence at the C-terminus are prenylated on the C-terminal cysteine (C), with a_1_ and a_2_ being aliphatic amino acids, although the a_1_ amino acid is more flexible. Proteins will be modified with a farnesyl moiety when X is an alanine (A), methionine (M), glutamine (Q), or serine (S). Proteins will be modified with a geranylgeranyl moiety when X is phenylalanine (F), isoleucine (I), or leucine (L). Additionally, RAB GTPases with a CC, CXC, CCX, CCXX, CCXXX, or CXXX C-terminal sequence will be modified with two geranylgeranyl moieties on the cysteines [28].

Furthermore, significantly enriched proteins in the combined PDE6D dataset were subjected to a STRING analysis in order to visualize protein clusters. The proteins that were not classified in a cluster based on the STRING analysis were subjected to a BioGRID interaction search. A GetGo analysis (min. count: 5) was performed for the Tier I and Tier II proteins of the UBL3 dataset, as described before [24].

### 2.6. Visible Immunoprecipitation (VIP) Assay

The eGFP-tagged PDE6D-encoding plasmid was co-transfected in HEK293T cells with plasmids encoding mRFP-tagged FAM219A WT/C > A, NIM1K WT/C > A, or UBL3 WT/C > A. The cells transfected with an empty mRFP vector (mRFP-control) or an empty eGFP vector (eGFP-control) were taken along to determine the background signal. After 40 h, the cells were lysed in lysis buffer (30 mM Tris, 150 mM NaCl, 0.5% NP40), protease inhibitor cocktail (Roche, 11697498001), 1% phosphatase inhibitor II (Sigma-Aldrich, P5726), and 1% phosphatase arrest III (G Biosciences, St. Louis, MO, USA, 786-452), and cleared by centrifugation at 14,000× *g* for 15 min at 4 °C. The cleared lysates were incubated with anti-GFP or anti-mCherry nanobodies prebound to glutathione-Sepharose beads (GE Healthcare, Eindhoven, The Netherlands, GE17-0756-01) as described by Katoh et al. [29] and incubated for 1.5 h at 4 °C. Subsequently, the beads were washed four times in lysis buffer, followed by a single wash with 1× PBS, and transferred to a 96-well imaging plate. The plates were stored at 4 °C until analysis and imaged on a Leica DMI6000B automated high-content microscope. For confirmation by western blot, beads were boiled in NuPAGE^TM^ LDS Sample Buffer (4×) with 50 mM DTT and loaded on a NuPAGE 4–12% BisTris gel. After transfer, membranes were blocked in 5% non-fat dry milk (Santa Cruz Biotechnology, Heidelberg, Germany, sc-2325) followed by overnight (ON) primary antibody incubation using mouse anti-GFP (Roche, 11814460001, 1:1000) and rabbit anti-RFP (Abcam, Amsterdam, The Netherlands, ab62341, 1:1000). Next, membranes were washed in 1× PBS + Tween-20 (0.2%) and incubated with secondary antibodies for 45 min. Finally, image acquisition was performed using an Odyssey DLx imager (LI-COR).

### 2.7. Yeast Two-Hybrid (Y2H) Assay

The plasmids encoding PDE6D fused to a DNA-binding domain (BD), and plasmids encoding FAM219A WT/C > A, NIM1K WT/C > A, or UBL3 WT/C > A fused to a transcription activation domain (AD) were transformed in yeast strains PJ69-4A and PJ69-4α respectively, and subsequently combined by yeast mating. The same procedure was performed with plasmids encoding PDE6D fused to AD and FAM219A WT/C > A, NIM1K WT/C > A, or UBL3 WT/C > A fused to BD. The diploids containing both plasmids were selected on media lacking leucine and tryptophan. In addition, the interactions were analyzed by assessment of reporter gene activation via growth on media additionally lacking histidine and/or adenine (*HIS3* and *ADE2* reporter gene), α-galactosidase (α-gal) colorimetric plate assays (*MEL1* reporter gene), and β-galactosidase (β-gal) colorimetric filter lift assays (*LacZ* reporter gene). Further, to the medium lacking leucine, tryptophan, and histidine (-LWH), 5 mM of the histidine biosynthesis inhibitor 3-amino-1,2,4-triazole (3-AT) was added to select for clones with high activation of the *HIS3* reporter gene. The following controls were used: pBD-PDE6D or pBD-UBL3 WT combined with pAD-WT (wild-type fragment C of lambda cl repressor) and pBD-WT combined with pAD-SV40 served as negative controls; pBD-WT combined with pAD-WT served as positive controls (Agilent Technologies, Amstelveen, The Netherlands).

### 2.8. Immunofluorescence Microscopy

The mIMCD3 cells transfected (25–30% transfection rate for all constructs) with eGFP-FAM219A WT/C > A, eGFP-NIM1K WT/C > A/S > K, or eGFP-UBL3 S > K, and mIMCD3 Flp-in cells stably expressing UBL3 WT or UBL3 C > A and tagged with eGFP were cultured on coverslips and subjected to serum deprivation (culturing medium with 0.2% FBS) to induce ciliogenesis. After 48 h, cells were fixed in 2% paraformaldehyde in PBS (pH 7.4), and washed twice with PBS. Cells were permeabilized for 5 min in 1% Triton X-100 (Sigma-Aldrich) in PBS, followed by two washes with PBS. In addition, the fixed coverslips were blocked at room temperature (RT) for 20 min in 2% Bovine Serum Albumin (BSA, Sigma-Aldrich, A6003) in PBS, followed by 1 h incubation with primary antibodies anti-pericentrin (Abcam, ab4448, 1:500) and anti-ARL13B (NeuroMab 75-287, 1:500) diluted in 2% BSA in PBS. Next, the coverslips were washed three times in PBS, followed by secondary antibody staining for 1 h at RT in 2% BSA in PBS. The cells were washed three more times in PBS and mounted onto microscopy slides using Vectashield with DAPI (Vector Laboratories, Newark, CA, USA, VECTH1200). The coverslips were analyzed on a Zeiss (Breda, The Netherlands) LSM880 or LSM900 laser scanning microscope with Airyscan using a 63× oil-immersion objective. Final images are depicted as maximum projections of Z-stacks.

In the assessment of protein localization, at least 15 transfected ciliated cells were analyzed for eGFP-FAM219A WT/C > A, eGFP-NIM1K WT/C > A, and UBL3 WT/C > A. In the assessment of localization of eGFP-NIM1K S > K and eGFP-UBL3 S > K, a total of 25 transfected ciliated cells were analyzed. Further, immunostainings were performed in duplicate on at least two biological replicates.

### 2.9. Animals

In the electroporation procedure, all live animal experiments were approved by the Massachusetts Eye and Ear Institutional Animal Care and Use Committees. This research followed the tenets of the ARVO Statement for the Use of Animals in Ophthalmic and Vision Research and the guidelines of the Massachusetts Eye and Ear Infirmary for Animal Care and Use and was specifically approved by the Institutional Animal Care and Use Committees at the Massachusetts Eye and Ear Infirmary. The wild-type CD-1 mice were purchased from Charles River Laboratories (Wilmington, MA, USA, stock # 022) and were used for UBL3 in vivo electroporation experiments. 

Furthermore, in the ultrastructure expansion microscopy procedure, C57BL/6J (B6) mice were handled in accordance with the statement of the “Animals in Research Committee” of the Association for Research in Vision and Ophthalmology, and experiments were approved by the Animal Ethics Committee (DEC) of the Radboud University Medical Centre (AVD10300 2016 758; RU-DEC-2016-0050). The mice were maintained at RT with a 12 h light/12 h dark cycle with lights on at 7:00 am and were fed ad libitum.

### 2.10. Sub-Retinal Injections and Electroporation

Neonatal P1 mice were placed on ice until fully anesthetized by hypothermia. When using a microscope to aid visualization, a small incision is made on the right eyelid, exposing the eye. Further, a small cut is then made at the limbus using a 30-gauge needle, and a Hamilton syringe with a 33-gauge blunt-ended needle was used to inject 0.5 µL purified plasmid DNA (eGFP-UBL3 WT/C > A) at 5 µg/µL into the subretinal space. The injected plasmid was electroporated into retinal cells using tweezer-type electrodes (Model 520, 7 mm diameter, BTX-Harvard Apparatus, Holliston, MA, USA), and five square 100 V pulses of 50 ms duration with 950 ms intervals were applied using a pulse generator (Model ECM 830, BTX-Harvard Apparatus). Following the injection, the eyes were treated with hydration gel and left on a heating pad to recover.

### 2.11. Dark and Light Adaptation

The light-adapted mice underwent 20 min of being exposed to 2000–3000 lux of light, while dark-adapted mice were housed overnight in a blackout cage before being sacrificed in complete darkness.

### 2.12. Retinal Immunohistochemistry

Three weeks post-injection, mice were sacrificed by CO_2_ under dark or light conditions, respectively. The eyes were dissected and placed in 4% paraformaldehyde (PFA; pH 7.4) in PBS for 10 min. The eye cups were then made by removing the cornea and placing it back in 4% PFA in PBS for 2 h. Subsequently, they were transferred to 30% sucrose in PBS overnight at 4 degrees. The next day, the eyecups were embedded in OCT and stored at −80 °C. The retinal cryosections were cut at 10 μm on ColorFrost slides. Further, the retinal sections were incubated with mouse anti-Rhodopsin antibody (Merck Millipore, Amsterdam, The Netherlands, MAB5356) in PBS overnight at 4 °C, followed by Alexa 555-conjugated secondary antibodies in PBS for 1 h and Hoechst (Thermo Scientific, 33342) staining for 10 min. Images were taken using the Eclipse Ti microscope (Nikon, Louvain, Belgium) and analyzed by NIS-elements Ar software.

### 2.13. Ultrastructure Expansion Microscopy (U-ExM) on Mouse Retina

The retinas were prepared for the U-ExM as earlier described [30]. In short, the eyes of P28 mice were enucleated and fixed for 15 min at RT in 4% PFA in PBS. Subsequently, the cornea and lens were removed with micro-scissors, followed by carefully removing the retina. The retinas were incised to flatten them and placed inside a 10 mm microwell of a 35 mm petri dish (P35G-1.5-10-C, MatTek, Ashland, MA, USA) for U-ExM processing. 

The expansion procedure was performed as earlier described [30]. In short, retinas were incubated ON in 100 μL of 2% acrylamide (AA; A4058, Sigma-Aldrich) + 1.4% formaldehyde (FA; F8775, Sigma-Aldrich) at 37˚C. The next day, retinas were incubated in 35 μL monomer solution (MS) composed of 25 μL of sodium acrylate (stock solution at 38% (*w*/*w*) diluted with nuclease-free water, 408220, Sigma-Aldrich), 12.5 μL of AA, 2.5 μL of N,N’-methylenbisacrylamide (BIS, 2%, M1533, Sigma-Aldrich), and 5 μL of 10× PBS for 90 min at RT. Subsequently, MS was removed, and retinas were incubated in 90 μL of MS with the addition of ammonium persulfate (APS, 17874, Thermo Fisher Scientific) and tetramethylethylenediamine (TEMED, 17919, Thermo Fisher Scientific) at a final concentration of 0.5% for 45 min at 4 °C followed by 3 h incubation at 37 °C. A 24-mm coverslip was added on top to close the chamber. Next, the coverslip was removed and 1 mL of denaturation buffer (200 mM SDS, 200 mM NaCl, 50 mM Tris Base in water (pH 9) was added into the MatTek dish for 15 min at RT with shaking to detach the gel from the dish. Afterwards, the gel was incubated in denaturation buffer for 1 h at 95 °C followed by ON at RT. The next day, the gel was sliced around the retina, which is still visible at this step, and expanded in three consecutive ddH_2_O baths. Next, the gel was manually sliced with a razorblade to obtain approximately 0.5 mm thick transversal sections of the retina to enable processing for immunostaining.

In immunostainings, the primary antibodies rabbit anti-UBL3 (ThermoFisher Scientific, PA5-96269, 1:250) and mouse tubulin AA344 (scFv-S11B, Beta-tubulin, 1:125) and AA345 (scFv-F2C, Alpha-tubulin, 1:125) [30] were incubated ON at 4˚C in 2% BSA in PBS. The image acquisition was performed on an inverted Leica (Wetzlar, Germany) Thunder DMi8 microscope using a 20× (0.40 NA) or 63× (1.4 NA) oil-immersion objective with Thunder SVCC (small volume computational clearing) mode at max resolution, adaptive as “Strategy,” and water as “Mounting Medium” to generate deconvolved images. The Z-stacks were acquired with 0.12 μm z-intervals and an x, y pixel size of 35 nm.

The expansion factor was calculated in a semiautomated way by measuring the full width at half maximum (FWHM) of the photoreceptor mother centriole proximal tubulin signal using the PickCentrioleDim plugin, as described before [30]. At least 10 photoreceptor mother centrioles FWHM were measured per replicate (n = 3) and compared to a pre-assessed value of U2OS centriole width (mean = 231.3 nm ± 15.6 nm) [30]. The calculation of the ratio between measured FWHM and known centriole width resulted in an expansion factor of 4.54. The scale bars shown in expansion microscopy images are corrected for the expansion factor.

## 3. Results

### 3.1. Tandem Affinity Proteomics of PDE6D Reveals a Strong Association to Prenylated (Ciliary) Proteins

In order to generate a comprehensive, high resolution interactome of PDE6D, we have performed tandem affinity purification (TAP), combining two affinity-purification steps to allow the isolation of high-purity PDE6D-bound protein complexes followed by MS analysis. This procedure was implemented on two different mammalian cell lines: HEK293T cells, transiently transfected with N-TAP-tagged PDE6D expressing plasmids, and mIMCD3 cells that stably express N-TAP-tagged PDE6D. In both datasets, we defined individual proteins as farnesylated, geranylgeranylated, or non-prenylated based on the C-terminal amino acid sequence of each protein. In both the HEK293T and mIMCD3 datasets, several known protein complexes were identified, including a geranylgeranylated guanine nucleotide-binding protein (G protein) complex, a geranylgeranylated SCF (SKP1-CUL1-F-box protein) ubiquitin ligase complex, and a non-prenylated COP9 signalosome (CSN) complex (Appendix A, indicated with *, ^#^, and ^, respectively), since these protein complexes function as one entity, we considered them as such in our prenylation quantification analysis (the quantification of prenylation percentage for individual proteins is shown in Appendix A and Appendix A). 

The analysis of the HEK293T cell dataset revealed a specific set of 26 proteins significantly associated with PDE6D (Figure 1A; Appendix A). Of these proteins, 16 (62%) are prenylated, whereof 10 contain a geranylgeranyl moiety and 6 contain a farnesyl moiety (Figure 1C, ‘HEK293T’).

The analysis of the mIMCD3 cell data revealed that 30 proteins are significantly associated with PDE6D (Figure 1B; Appendix A). Based on the Ca_1_a_2_X criteria, 15 of the 30 proteins (50%) are prenylated, whereof 9 contain a geranylgeranyl moiety and 6 contain a farnesyl moiety (Figure 1C, ‘mIMCD3’).

Combining both datasets resulted in a set of 42 unique proteins (Appendix A), of which a total of 21 proteins are prenylated (50%), including 14 geranylgeranylated proteins and 7 farnesylated proteins (Figure 1C, ‘Combined’). The STRING analysis of the combined dataset shows that the PDE6D-associated proteins are tightly interconnected and can be divided into different clusters based on their function (Figure 1E). These clusters include: photoreceptor integrity; GTPases, subdivided into G proteins and small GTPases, including their regulators; nuclear protein import; ubiquitination; and miscellaneous.

In order to validate that the identification of prenylated proteins is specific for PDE6D, we performed an identical prenylation quantification analysis for potential RAF1 (control) interactors in both the HEK293T and the mIMCD3 datasets. The analysis of the HEK293T dataset revealed 132 potential RAF1 interactors, whereof only 4 (3%) proteins are prenylated, including 1 geranylgeranylated protein and 3 farnesylated proteins (Figure 1C, ‘RAF1 HEK293T’; Appendix A). The analysis of the mIMCD3 dataset revealed 113 potential RAF1 interactors, whereof only 2 (2%) were prenylated proteins, including 1 geranylgeranylated and 1 farnesylated protein (Figure 1C, ‘RAF1 mIMCD3; Appendix A).

In order to determine the most conserved interactors of PDE6D, we compared both datasets (Appendix A). A total of 14 proteins are present in both datasets, whereof 11 proteins are prenylated (79%), including 6 geranylgeranylated and 5 farnesylated proteins (Figure 1C, ‘Overlap’). Two of the three non-prenylated proteins are ARL2 and ARL3, which play a fundamental role in prenylated cargo release from PDE6D [9].

In both datasets, we noticed a separation between proteins with a label-free quantification (LFQ) intensity score, indicating the relative number of proteins in the dataset, above and below 30 (Figure 1A,B, dashed line). The expectation was that proteins with an LFQ intensity score above 30 would belong to the strongest interactors of PDE6D. In total, 15 proteins have an LFQ intensity score above 30 in at least one dataset (Appendix A). Out of these 15 proteins, 12 are prenylated (80%), whereof six are geranylgeranylated and six farnesylated (Figure 1C, ‘Int. > 30 either’).

In total, 8 proteins have an LFQ intensity score above 30 in both datasets (Appendix A), whereof 7 proteins are prenylated (88%), including 3 geranylgeranylated and 4 farnesylated (Figure 1C, ‘Int. > 30 both’). The protein that is not prenylated is ARL3, which plays an important role in the trafficking process of prenylated proteins [9], as indicated before.

In addition, within the prenylated protein sets, a distinction can be made between high affinity binding and low affinity binding cargo proteins based on the amino acid at the −3-position relative to the prenylated cysteine at the C-terminus of the cargo protein, as proposed before [31]. More specifically, a serine at the −3-position results in a stronger binding affinity to PDE6D. Furthermore, a serine at the −3-position seems to be conserved in ciliary cargo proteins [31]. Therefore, we examined the C-terminal protein sequence of the prenylated proteins in our datasets to determine the high-affinity binding proteins containing a serine at the −3-position. Out of the 25 prenylated proteins in the combined dataset, 12 proteins (48%) contain a serine at the −3-position, whereof 7 are geranylgeranylated and 5 are farnesylated (Figure 1D). For at least three of the proteins in our dataset, including FAM219A, Nim1k, and UBL3, a functional role in cilia has not yet been proposed.

### 3.2. PDE6D Strongly Interacts with Prenylated FAM219A, NIM1K, and UBL3

The proteins FAM219A, NIM1K, and UBL3 all contain a serine at the −3-position relative to the prenylated cysteine (Figure 1D), suggesting a strong interaction of these prenylated proteins with PDE6D [31].

We investigated whether PDE6D is directly interacting with FAM219A, NIM1K, and UBL3 and whether this interaction is dependent on the prenylation moiety of these proteins. Therefore, we performed a VIP assay to investigate the interaction of PDE6D with the prenylated wild-type (WT) forms of FAM219A, NIM1K, and UBL3 and the non-prenylated mutant (C > A at the -4-position of the C-terminus) forms of these proteins. We found that PDE6D is interacting with all three prenylated WT proteins, but not with the non-prenylated C > A form of these proteins (Figure 2A and Appendix A). The prenylation-dependent direct interaction of PDE6D with FAM219A, NIM1K, and UBL3 is further confirmed by a Y2H assay (Figure 2B). For the interaction of PDE6D and FAM219A WT, activation of all reporter genes for growth, α-gal, and β-gal assays could be observed. Although none of the reporter genes are activated for the combination of pBD-PDE6D and pAD-FAM219A C > A or the negative control, suggesting a valid interaction, autoactivation could be observed when the tags were switched. Further, for the interactions of PDE6D with NIM1K WT and UBL3 WT, activation of all reporter genes for growth, α-gal, and β-gal assays could be observed, while none of the reporter genes are activated for the combination of PDE6D with NIM1K C > A and UBL3 C > A.

### 3.3. Ciliary Proteins FAM219A, NIM1K, and UBL3 Localize in a Prenylation-Dependent Manner

Due to the fact that PDE6D is involved in the trafficking of cargo proteins to the cilium and a serine at the −3-position relative to the prenylated cysteine is proposed to be conserved in ciliary proteins [31], we hypothesized that prenylated FAM219A, NIM1K, and UBL3 localize to the cilium. In order to investigate this, we transfected eGFP-FAM219A WT/C > A and eGFP-NIM1K WT/C > A into mIMCD3 cells. Furthermore, we generated mIMCD3 Flp-in cells either stably expressing eGFP-tagged UBL3 WT or eGFP-tagged UBL3 C > A. Both NIM1K WT (Figure 3B, upper panel) and UBL3 WT (Figure 3C, upper panel; Appendix A) localize inside the cilium over its full length, with UBL3 WT also detected in vesicle-like structures around the base of the cilium (Figure 3C, upper panel). The prenylation mutants of NIM1K (Figure 3B, middle panel) and UBL3 (Figure 3C, middle panel) localize to the base of the cilium and to centrosomes. We could not detect FAM219A WT inside the cilium (Figure 3A, upper panel). However, FAM219A C > A localizes to the base of the cilium and to centrosomes (Figure 3A, bottom panel), identical to the prenylation mutants of NIM1K and UBL3.

Additionally, since the proteins that contain a serine at the −3-position relative to the prenylated cysteine are proposed to be ciliary proteins, and proteins that contain a lysine at this −3-position are proposed as non-ciliary proteins [31], we changed this serine in NIM1K and UBL3 to a lysine (S > K) to investigate the effect on ciliary localization. Our results show that both eGFP-NIM1K S > K (Figure 3B, bottom panel, white arrow) and eGFP-UBL3 S > K (Figure 3C, bottom panel) are still able to localize inside the cilium, in 56% (14/25 cilia) and 100% of transfected ciliated cells, respectively. Moreover, eGFP-NIM1K S > K could also be prominently detected at the basal bodies in 8% (2/25 cilia) of transfected ciliated cells (Figure 3B, bottom panel, white arrowhead). In 36% (9/25 cilia) of transfected ciliated cells, we observed both a prominent localization at the basal bodies and a faint localization in the cilium (Appendix A).

### 3.4. Affinity Proteomics of UBL3 Reveals Its Association to sEVs and Ciliogenesis

Since the LFQ intensity score of UBL3 is above 30 in both datasets and UBL3 is the most significant protein in the HEK293T dataset (based on a *t*-test; *p*-value = 2.55 × 10^−13^; Appendix A), we decided to perform affinity proteomics with this protein to gain further insight in the association of UBL3 with cilia and vesicle-like structures.

In this case, we performed affinity proteomics on cilia-induced mIMCD3 Flp-in cells, stably expressing either N-TAP-tagged UBL3 WT or UBL3 C > A. The proteins that were enriched in UBL3 WT samples compared to both UBL3 C > A and RAF1 (control) were considered potential UBL3 interactors (Figure 4A and Appendix A). The potential UBL3 interactors are divided into Tier I (green) and Tier II (orange), based on the significant enrichment score (Tier I: *t*-test FDR ≤ 0.05 + Log2 ratio ≥ 5; Tier II: *t*-test *p* ≤ 0.05 + Log2 ratio ≥ 5). Tier I contains six proteins that are most strongly associated with UBL3, including PDE6D, confirming the strong prenylation-dependent interaction of PDE6D with UBL3.

The GetGo analysis of the Tier I and Tier II proteins (Figure 4B; Appendix A), which contain a total of 21 proteins, showed that these proteins are mainly present in the cytosol (12/21 = 57%) and/or in extracellular exosomes (11/21 = 52%), a type of small extracellular vesicles. Furthermore, seven proteins (33%) are involved in protein metabolism, and six proteins (29%) are involved in disease. In addition, four of the disease-related proteins do also have an effect on ciliogenesis, as indicated in the category ‘Syscilia-siRNA.’ A total of 8 out of the 21 proteins (38%) are indicated in this category, meaning that ciliogenesis is affected when these proteins are downregulated by siRNAs [24].

### 3.5. UBL3 Localizes to Specific Photoreceptor Compartments

Several PDE6D cargos play important roles in photoreceptor function [14,15,16,17], which are also detected in our datasets (Figure 1). In order to elucidate whether UBL3 also plays a role in photoreceptors, we electroporated cDNA plasmids of eGFP-UBL3 WT and eGFP-UBL3 C > A into P1 mouse retinas. We observed that eGFP-UBL3 WT is mainly expressed in the outer segments of photoreceptors, with some signal being observed in the inner segments, ONL, and synapse regions when recombinant protein expression is high (Figure 5A, upper and middle panels). In contrast, eGFP-UBL3 C > A mislocalized to the inner segments and ONL, with very little signal reaching the outer segments of photoreceptors (Figure 5A, bottom panel). Further, there were no obvious differences between light- and dark-adapted retinas; both eGFP-UBL3 WT and eGFP-UBL3 C > A localized in their respective locations, regardless of light exposure (Figure 5A).

Additionally, in order to further elucidate the subcellular localization of UBL3 inside the photoreceptor, we performed U-ExM on mouse retinas, followed by UBL3 and tubulin antibody staining. Based on the tubulin staining, we could observe the previously described photoreceptor cilium regions, including the basal body (BB) and daughter centriole (DC), the connecting cilium (CC), which bridges the inner segment (IS) to the outer segment (OS), and the bulge region (BR), the base of the OS [30]. Based on these regions, we could show that UBL3 accumulates in the distal part of the BR and extends into the distal axoneme, the spine of the OS (Figure 5B,C). Furthermore, UBL3 could also be detected inside the CC (Figure 5B,C).

## 4. Discussion

In order to gain more insights into the PDE6D-associated disease mechanisms, we set out to dissect the conserved interactome of PDE6D by performing tandem affinity purification (TAP) in two different mammalian cell lines, followed by MS analysis with label-free quantification. By this approach, we have identified a specific set of PDE6D-interacting proteins that are known to be involved in photoreceptor integrity, GTPase activity, nuclear import, or ubiquitination. The majority of the strongest interactors of PDE6D are prenylated, up to 79% for overlapping proteins (Figure 1C) in both datasets and even up to 88% for proteins with an intensity score above 30 (Figure 1C). The small GTP-binding proteins ARL2 and ARL3 were also identified among the strongest interactors of PDE6D. These proteins function as dissociation factors to release the prenylated cargo from PDE6D by inducing a conformational change in the hydrophobic pocket, squeezing out the prenylation moiety [9]. Although the mechanism of cargo release from PDE6D is similar for ARL2 and ARL3, both act on different cargos. The ARL2 displaces low-affinity cargo, whilst the ARL3 dissociates high-affinity cargo by partly folding into the hydrophobic pocket of PDE6D [9,32]. Similarly, our data shows a stronger association of PDE6D with ARL3 compared to ARL2. Taken together, almost all of the strongest interactors of PDE6D are prenylated or play a fundamental role in the prenylation trafficking process, confirming not only the importance of PDE6D in prenylated cargo trafficking but also the specificity of our approach.

Furthermore, by focusing on the prenylated proteins, we could determine the distribution between farnesylated and geranylgeranylated proteins. Overall, more geranylgeranylated proteins were identified compared to farnesylated proteins, which could indicate that the binding affinity of PDE6D to a geranylgeranyl moiety is stronger compared to a farnesyl moiety. This is in accordance with previous findings showing that the geranylgeranyl moiety is bound deeper in the hydrophobic pocket of PDE6D compared to the farnesyl moiety, resulting in more hydrophobic contacts and thus increased binding affinity [31]. For the interactors of PDE6D, with an LFQ intensity score above 30, the distribution of geranylgeranylated and farnesylated proteins is almost equal, so there are relatively more farnesylated proteins in this group compared to the overall dataset. It could be speculated that the difference in binding affinity for both prenylation moieties is neutralized by additional binding sites at the cargo protein, as shown for RPGR and speculated for GRK7 [5,33,34]. Moreover, the amino acid at the −3-position relative to the prenylated cysteine at the C-terminus of the cargo protein also has a strong influence on the affinity for PDE6D. A serine on the −3-position results in an additional hydrogen bond between the prenylation moiety and PDE6D, while this is not the case for a lysine at the −3-position [31]. Interestingly, a serine at the −3-position seems to be conserved in ciliary cargo proteins [31]. This theory is in accordance with our data, since almost all proteins containing a serine at the −3-position in our datasets (Figure 1D) are shown to have a role in cilia regulation, including photoreceptor cilia [16,35,36,37,38,39,40,41,42,43,44,45,46,47,48,49,50,51].

In the case of proteins FAM219A, NIM1K, and UBL3, however, a functional role in cilia has not yet been described to date, although UBL3 is upregulated in CEP290-mutated patient-derived iPSC-RPE [52]. We identified that PDE6D interactors NIM1K and UBL3 localize inside the cilium in a prenylation-dependent manner, while prenylated FAM219A was not detected inside the cilium. Interestingly, the non-prenylated form of FAM219A did localize to the base of the cilium, identical to the prenylation mutants of ciliary proteins NIM1K and UBL3, thus still suggesting a ciliary role for FAM219A. The prenylation-dependent localization is in line with previous studies showing that PDE6D is necessary for extraction of prenylated proteins from the donor membrane, for association of prenylated cargo with intraflagellar transport trains, and for enrichment of prenylated proteins in the photoreceptor outer segment, while it is dispensable for ciliary targeting of prenylated proteins [39,46,53,54,55,56]. We demonstrate that NIM1K and UBL3 are novel ciliary proteins, in line with the proposed theory that a serine at the −3-position relative to the prenylated cysteine is conserved for ciliary proteins. We showed that NIM1K and UBL3 are still able to localize to the cilium with a lysine at the −3-position relative to the prenylated cysteine. Since a serine results in a stronger binding affinity to PDE6D [31], it could be speculated that a serine is more effective for ciliary targeting compared to a lysine. This is in accordance with our finding that NIM1K S > K was also detected at the basal bodies. In order to further clarify the effect of a serine to lysine change on ciliary localization, assessment of endogenously expressed proteins is necessary. In addition to the prenylation domain and the serine at the −3-position relative to the prenylated cysteine, other domains could influence the ciliary targeting, as recently shown for INPP5E [57]. However, this is less likely for FAM219A, NIM1K, and UBL3, since removal of the prenylation moiety results solely in basal body localization, while non-prenylated INPP5E still localizes inside the cilium.

In the case of the localization of UBL3, we also observed that it localizes to vesicle-like structures around the base of the cilium. Similarly, we found that UBL3 is associated with proteins involved in ciliogenesis and extracellular exosomes, a type of small extracellular vesicles (sEVs). However, whether the UBL3 containing vesicle-like structures around the base of the cilium are indeed exosomes remains to be determined. 

In a non-ciliary context, UBL3 has been shown to be involved in protein sorting by sEVs. Although a strong association with exosomes was shown, a role in other sEVs cannot be ruled out [19]. Moreover, distinguishing different forms of sEVs is rather complex due to overlap in composition, content, localization, and size between different sEVs [58]. The lack of glycolytic enzymes and cytoskeletal proteins are proposed as negative markers for exosomes [59]. In our proteomics data, we found UBL3 associated with Ogdh (Tier II), a protein involved in glycolysis, and Dctn4 (Tier II), a cytoskeleton-associated protein, indicating a role for UBL3 in non-exosomal sEVs. In contrast, UBL3 is strongly associated with several exosomal-associated (Tier I) proteins, including Ddr1, Mtch2, Sdcbp, and Vdac2, as revealed by our GetGo analysis. However, Sdcbp (Tier I), together with Pdcd6ip (Tier II), regulates exosome biogenesis [60]. Although it is challenging to pinpoint UBL3 to a specific form of sEVs, a functional role for UBL3 in sEVs is highly plausible.

In recent years, studies showing the importance of sEVs in cilia biology have been accumulating. Ciliary sEVs play a role in signaling [61,62,63], ciliary disassembly and resorption [64,65], and photoreceptor outer segment morphogenesis [18]. The signaling role of ciliary sEVs is in line with our proteomics data showing a strong association of UBL3 with signalling protein Notch2 (Tier I). Notch2 has been shown to play a role in signaling via sEVs [66]. Furthermore, our GetGo analysis revealed that Notch2 is one of the proteins affecting ciliogenesis when downregulated, which is supported by findings showing that Notch signaling controls cilia length during renal tubule morphogenesis [67]. Although Notch signaling has not yet been shown to be directly associated with photoreceptor outer segment morphogenesis, it does play an important role in photoreceptor formation by regulating photoreceptor fate during retinal development [68,69]. 

Another UBL3-associated protein involved in signaling, TGFBR2 (Tier II), is also implicated in ciliogenesis, as revealed by our GetGo analysis, and associated with exosome-dependent signaling [70], thereby further establishing the connection between cilia, sEVs, and signaling.

Taken together, our results suggest a strong association of UBL3 with ciliary sEVs. Combined with previous findings describing a role of UBL3 in protein sorting to sEVs [19], we propose that UBL3 may play a similar role in protein sorting to ciliary sEVs. Since the previously described role of UBL3 in protein sorting to sEVs was investigated in a non-ciliary context, it could be speculated that these sEVs are ciliary sEVs.

Furthermore, since PDE6D interacts strongly with geranylgeranylated UBL3, as further confirmed by our affinity proteomics of UBL3, it could be hypothesized that PDE6D is also involved in the regulation of ciliary sEVs. The RAB GTPases, which play important roles in ciliary vesicle trafficking [44], are trafficked by PDE6D [71], as also confirmed by our data, indicating an indirect regulatory role of PDE6D in ciliary vesicle transport. Although the nature of these RAB-mediated ciliary vesicles remains to be elucidated, one of the RAB GTPases detected in our PDE6D datasets, RAB28, is involved in regulating extracellular vesicle shedding at the periciliary membrane [46]. A similar process is described in murine photoreceptors, in which knock-out of *Rab28* leads to elongated and enlarged outer segments, suggesting a disc shedding impairment, explaining the cone-rod dystrophy caused by *RAB28* mutations in humans [72]. The active peripherin-dependent suppression of extracellular vesicle shedding in photoreceptors further highlights the importance of this process in outer segment formation [18]. 

Our results show that UBL3 localizes to the bulge region of photoreceptors in a prenylation-dependent manner. Based on this specific localization combined with the proposed function of UBL3 in sorting of proteins to ciliary sEVs, an important process in photoreceptor outer segment formation [18], we suggest a potential role for UBL3 in sorting of proteins towards the photoreceptor outer segment. Since no obvious phenotype has been previously described for UBL3 knock-out in mice [19], further assessment of retinal tissue from these mice could elucidate the precise function of UBL3 in photoreceptors. Moreover, to find out whether FAM219A and NIM1K also play a role in photoreceptors, these proteins should be further studied in the context of the retina.

## Figures and Tables

**Figure 1 cells-12-00312-f001:**
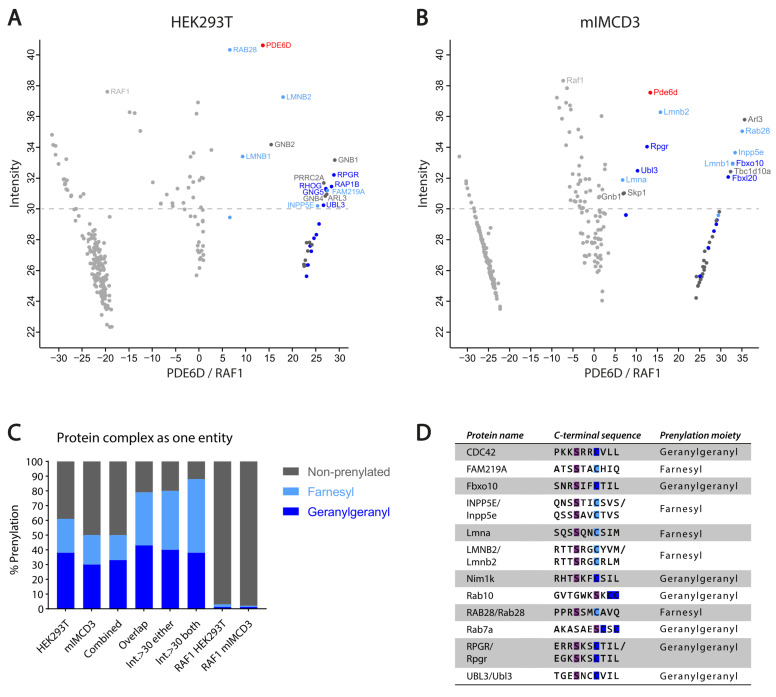
Identification and clustering of potential PDE6D interactors. (**A**) Scatterplot showing enriched proteins comparing PDE6D to control (RAF1) in HEK293T cells. The bait protein (PDE6D) is shown in red. Significantly enriched geranylgeranylated proteins are indicated in dark blue and significantly enriched farnesylated proteins are indicated in light blue. Significantly enriched non-prenylated proteins are indicated in dark grey. Proteins with an LFQ intensity value above 30 (indicated with dashed line) are labeled. X-axis represents log2 ratio between PDE6D and RAF1 (control). Y-axis represents label-free quantification (LFQ) intensity score, indicating the relative number of proteins in the dataset. (**B**) Scatterplot showing enriched proteins comparing PDE6D to control (RAF1) in mIMCD3 Flp-in cells. Indicated colors and axis are identical to A. (**C**) Potential PDE6D interactors categorized based on prenylation moiety in different (sub)datasets, with protein complexes considered as one entity. Bars indicated with ‘HEK293T’ and ‘mIMCD3’ represents individual datasets performed in HEK293T and mIMCD3 Flp-in cells, respectively. Bar indicated with ‘Combined’ represents the complete set of unique proteins (including proteins present in both datasets) resulting from combining both datasets. Bar indicated with ‘Overlap’ represents all proteins that are present in both datasets. Bars indicated with ‘Int. > 30’ represent proteins that have an LFQ intensity score above 30 in at least one dataset (‘either’) or in both datasets (‘both’). Bars indicated with ‘RAF1 HEK293T’ and ‘RAF1 mIMCD3’ show the prenylation percentage in the control. Percentage of geranylgeranylated proteins is indicated in dark blue, percentage of farnesylated proteins is indicated in light blue, and percentage of non-prenylated proteins is indicated in dark grey. (**D**) Overview of the prenylated proteins from both the HEK293T and mIMCD3 dataset containing a serine (indicated in purple) at the −3-position relative to the prenylated cysteine (indicated in blue). (**E**) Representation of the STRING analysis of potential PDE6D interactors subdivided in different clusters. Protein labels are colored based on prenylation moiety of the protein corresponding to (**A**–**D**), with geranylgeranyl in dark blue, farnesyl in light blue, and non-prenylated in dark grey.

**Figure 2 cells-12-00312-f002:**
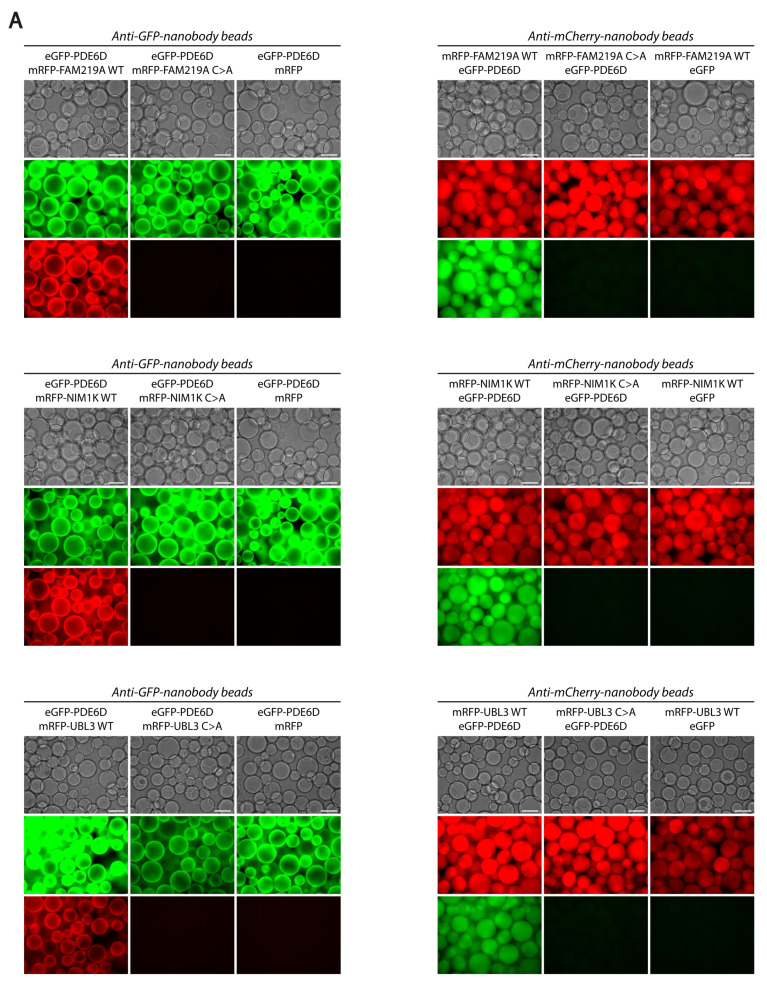
Confirmation of prenylation-dependent PDE6D interactions by visible immunoprecipitation (VIP) assays and yeast two-hybrid (Y2H) assays. (**A**) Results of VIP assays using anti-GFP-nanobody beads (**left**) and anti-mCherry-nanobody beads (**right**). Co-immunoprecipitation from HEK293T cells was performed using eGFP-tagged PDE6D and mRFP-tagged prenylated WT versus non-prenylated C > A proteins, including FAM219A WT/C > A, NIM1K WT/C > A, and UBL3 WT/C > A. Scale bar: 100 μm (**B**) Results of Y2H assays using PDE6D, FAM219A WT/C > A, NIM1K WT/C > A, and UBL3 WT/C > A as baits and preys. Interactions were analyzed by assessment of reporter gene activation via growth on two different media additionally lacking histidine and/or adenine (-LWH + 3-AT and -LWHA), α-galactosidase (α-gal) colorimetric plate assays (*MEL1* reporter gene), and β-galactosidase (β-gal) colorimetric filter lift assays.

**Figure 3 cells-12-00312-f003:**
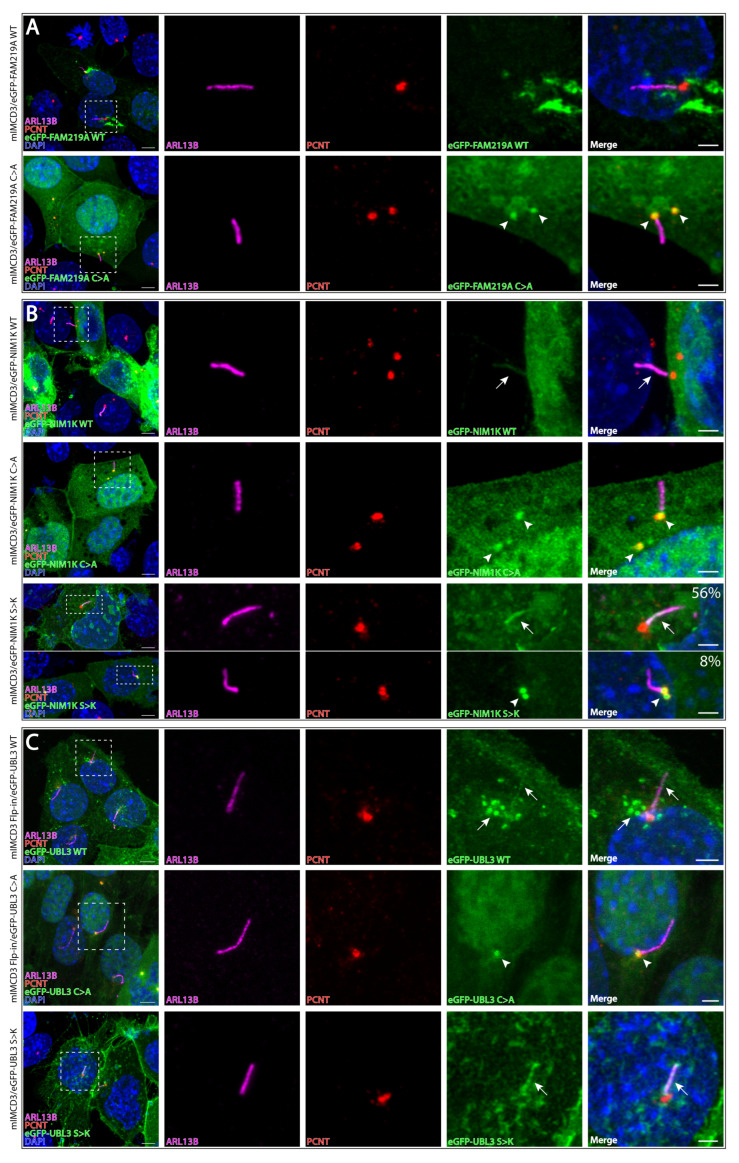
Comparative localization analysis of eGFP-FAM219A WT/C > A, eGFP-NIM1K WT/C > A/S > K, and eGFP-UBL3 WT/C > A/S > K expressed in mIMCD3 cells. (**A**) Ciliogenesis induced mIMCD3 cells transfected with eGFP-FAM219A WT (upper panel) and eGFP-FAM219A C > A (bottom panel). Arrowheads indicate eGFP-FAM219A C > A staining at the basal bodies/centrosomes. (**a-c**) Cilia were stained with antibodies against ARL13B (magenta). Basal bodies were stained with antibodies against pericentrin (PCTN, red). Nuclei were stained with DAPI (blue). (**B**) Ciliogenesis induced mIMCD3 cells transfected with eGFP-NIM1K WT (upper panel), eGFP-FAM219A C > A (middle panel), and eGFP-NIM1K S > K (bottom panel). Arrows indicate eGFP-NIM1K WT and eGFP-NIM1K S > K staining inside the cilium. Arrowheads indicate eGFP-NIM1K C > A and eGFP-NIM1K S > K staining at the basal bodies/centrosomes. Percentages indicated in bottom panel show the percentage of transfected ciliated cells that either show a ciliary localization or a basal body localization. (**C**) Ciliogenesis induced mIMCD3 cells stably expressing eGFP-UBL3 WT (upper panel) and eGFP-UBL3 C > A (middle panel), and transiently expressing eGFP-UBL3 S > K (bottom panel). Arrows indicate eGFP-UBL3 WT staining inside the cilium and in vesicle-like structures around the base of the cilium, and eGFP-UBL3 S > K staining inside the cilium. Arrowheads indicate eGFP-NIM1K C > A staining at the basal bodies/centrosomes. Scale bar: 5 μm (inset: 2 μm).

**Figure 4 cells-12-00312-f004:**
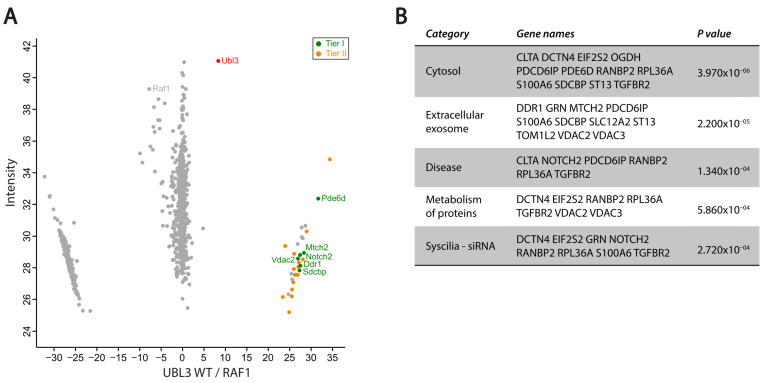
Identification and clustering of potential UBL3 interactors. (**A**) Scatterplot showing enriched proteins comparing UBL3 WT to control (RAF1) in mIMCD3 Flp-in cells. The bait protein (UBL3) is shown in red. Significantly enriched proteins (UBL3 WT vs. RAF1 + UBL3 C > A) are indicated in green (Tier I) and orange (Tier II), with Tier I and Tier II proteins determined by FDR ≤ 0.05 and *p* ≤ 0.05, respectively. X-axis represents log2 ratio between UBL3 WT and RAF1 (control). Y-axis represents label-free quantification (LFQ) intensity score, indicating the relative number of proteins in the dataset. (**B**) Summary of GetGo analysis of potential UBL3 interactors, including Tier I and Tier II proteins. Original data is listed in Appendix A.

**Figure 5 cells-12-00312-f005:**
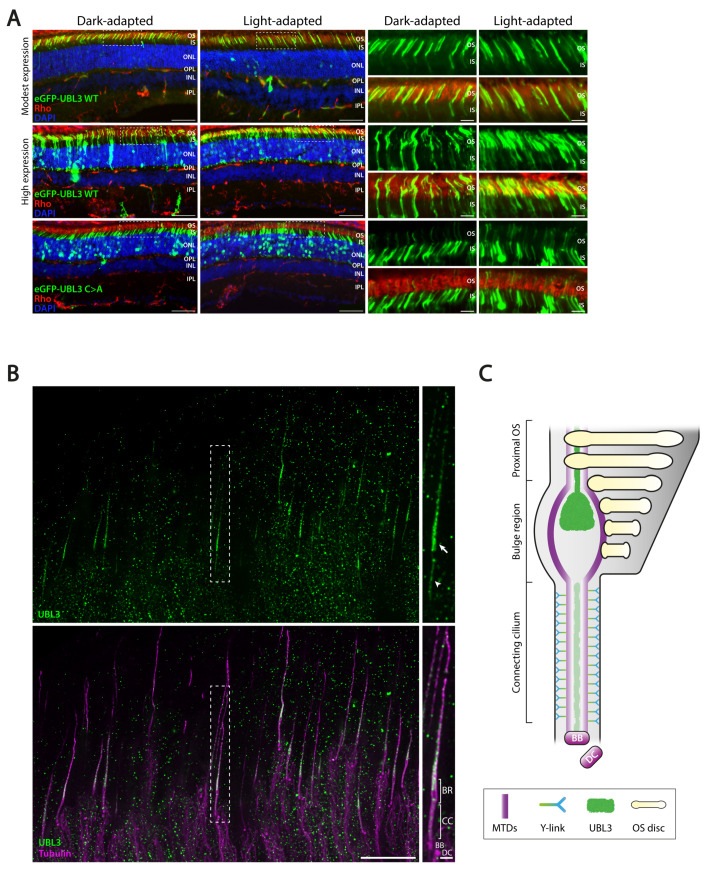
Localization of UBL3 inside the mouse photoreceptor. (**A**) Dark- and light-adapted mouse retinas expressing eGFP-UBL3 WT (modest vs. high expression) or eGFP-UBL3 C > A. Retinas were stained for rhodopsin (red) and DAPI (blue). Abbreviations: OS, outer segment; IS, inner segment; ONL, outer nuclear layer; OPL, outer plexiform layer; INL, inner nuclear layer; IPL, inner plexiform layer. Scale bar: 50 μm (inset: 10 μm). (**B**) U-ExM image of P28 mouse photoreceptors stained for UBL3 (green) and tubulin (purple). UBL3 localization in the bulge region (BR) and in the distal axoneme is indicated with a white arrow. UBL3 localization inside the connecting cilium (CC) is indicated with a white arrowhead. Abbreviations: BR, bulge region; CC, connecting cilium; MC, mother centriole; DC, daughter centriole. Scale bar: 5 µm (inset: 500 nm). (**C**) Schematic representation of a part of a wild-type (WT) photoreceptor consisting of the connecting cilium (CC), the bulge region (BR), and the proximal outer segment (OS), including its distal axoneme and membranous stacked discs. The microtubule doublets (MTDs), forming the axoneme, are built-up from the basal body (BB), accompanied by the daughter centriole (DC). MTDs in the CC are connected to the membrane by Y-links. UBL3 localizes to the distal part of the bulge region, where MTDs are more dispersed and Y-links are absent, extending inside the distal axoneme and to a lesser extent in the CC, as shown in (**B**).

**Table 1 cells-12-00312-t001:** Overview of Gateway destination vectors.

Plasmid	N-Tag	Purpose	Reference
pBD-GAL4_Cam/DEST	pBD	DNA-binding domain Y2H	[20]
pAD-GAL4_2.1/DEST	pAD	Activating domain Y2H	[20]
SF-TAP/N-TAP	Tandem StrepII/FLAG	Tandem affinity Purification	[21]
pcDNA5/FRT/TO GFP	eGFP	Immunofluorescence/VIP assay/electroporation	[22]
pDest-733	mRFP	Immunofluorescence/VIP assay	[23]

## Data Availability

All data from this study is included in the results and Appendix A. Raw data can be obtained from the authors upon reasonable request.

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
