# Peer review of "PDE6D Mediates Trafficking of Prenylated Proteins NIM1K and UBL3 to Primary Cilia"

_cells, 2023, doi:10.3390/cells12020312_

Round 1
Reviewer 1 Report
The manuscript by Faber et al describes PDE6D mediated trafficking of prenylated proteins NIM1K and 2 UBL3 to primary cilia. The authors purified the prenyl-18 ated cargo of PDE6D by affinity proteomics to gain insight into the PDE6D-associated proteins, including FAM219A, NIM1K, and ubiquitin-like protein 3 (UBL3). The authors show that NIM1K and UBL3 localize to cilia and/or photoreceptor. The authors propose that UBL3 might play a role in PDE6D-associated retinal degeneration. Overall, the authors have done an impressive set of experiments to define novel cargos of PDE6D. However, I have a few comments on the data interpretation and presentation, addressing which might improve the manuscript.
First, the localization of FAM219A, NIM1K, and UBL3 in primary cilia of IMCD3 cells is not clear, at least in the images provided, unless some of them are mutated (S>K). Paradoxically, the S>K mutation enhances the ciliary localization, something that needs to be addressed by the authors if possible. Quantification of this data along with studying prenylation mutants in IMCD3 cells would improve this whole dataset.
Second, the localization of UBL3 as tested by electroporation in photoreceptors is very interesting. Based on the images provided with co-localization marker Rhodopsin, I am not sure how the authors surmise on the localization of UBL3 as shown in Fig. 5C.
Third, did the authors test any of the other PDE6D interacting proteins in the photoreceptors, especially as UBL3 knockout has no reported eye phenotypes?
Minor points:
Image 4A, please check green/merge panels, as the green image looks truncated.
Reviewer 2 Report
Ciliopathies are deadly human diseases caused by dysfunctional cilia. Until now, a curative therapy is not available. Accordingly, the uncovering of molecular mechanisms underlying the development of ciliopathies is of great importance. In this study, Faber and colleagues aimed to get more insights into the interactome and functional network of the ciliopathy protein PDE6D. By performing a TAP assay, they found a strong association between PDE6D and FAM219A, NIM1K as well as UBL3. Further investigations (VIP assay, Y2H assay) revealed that PDE6D directly interacts with FAM219A, NIM1K and UBL3 in a prenylation-dependent manner. Since PDE6D is involved in trafficking of cargo proteins to cilia, Faber et al. investigated the subcellular localization of wild-type (WT) FAM219A, NIM1K and UBL3 as well as of mutant (MUT; non-prenylated) FAM219A, NIM1K and UBL3. The authors detected a prenylation-dependent ciliary localization of the three proteins. Following the significance evaluation of the score data, Faber and colleagues focussed on UBL3 for further analyses. Affinity proteomics points to a link between UBL3 and ciliogenesis as well as to a link between UBL3 and extracellular vesicles (EVs). Lastly, the authors referred to the important function of other PDE6D cargos in photoreceptor function and aimed to clarify whether UBL3 is also an important player in photoreceptors. For this purpose, they investigated the localization of UBL3 WT and UBL3 MUT in mouse photoreceptors after electroporation. While UBL3 WT localized mainly to the outer segments of photoreceptors, UBL3 MUT was mostly detected in the inner segments of the photoreceptors. By using ultrastructure expansion microscopy, Faber et al. detected UBL3 inside the connecting cilium and in the distal part of the bulge region.
The topic analysed in this study is of current importance and of interest to a broad readership. Most of the studies provided in this manuscript were carefully performed and are presented in a comprehensible manner. Nevertheless, there are several points which have to be improved before the manuscript is suitable for publication in Cells:
1) Line 209: Faber and colleagues should specify what pBD-WT and pAD-WT exactly are.
2) Line 243: “feed” should be revised into “fed”.
3) Line 270: “Hoescht” should be corrected to “Hoechst”
4) The authors mentioned that Pde6d-/- mice solely exhibit reduced body weight and recessive rod-cone dystrophy implicating an essential function of PDE6D in rod and cone development and/or maintenance. Nevertheless Faber et al. analyzed the interactome of PDE6D by performing TAP in HEK293T cells and in mIMCD3 cells instead using a photoreceptor cell line such as the 661W cell line. This selection should be explained.
5) Faber and colleagues wrote: “For at least three of the proteins in our dataset, including FAM219A, Nim1k, and UBL3, a potential role in cilia has not yet been proposed to date.” (lines 378-380). In case of UBL3, this statement needs to be modified, as UBL3 was found to be altered in ciliopathy patient-derived iPSC-RPE cells (May-Simera et al., 2018, Cell Rep., DOI: 10.1016/j.celrep.2017.12.038).
6) Referring to the localization studies, any statistics are missing. The transfection rate should be mentioned as well as the number of cilia, basal bodies and/or centrosomes at which FAM219A MUT, NIM1K WT/MUT and UBL3 WT/MUT are present.
7) Regarding the affinity proteomics study: Why was RAF1 considered as a suitable control? This choice should be explained.
8) The first part of the discussion (about PDE6D, FAM219A, NIM1K and UBL3) contains too many repetitions and should be shortened.
9) A recent report shows the regulation of WNT signaling by sEVs (Volz et al., 2021, Nat. Comm., DOI: 10.1038/s41467-021-25929-1). This citation should be added to the references.
10) As the interactome of PDE6D was investigated by using HEK293T cells and mIMCD3 cells, the question is obvious whether endogenous UBL3 is localized in other cells than photoreceptors and – if that is so – whether UBL3 is present in cilia of other cells.
11) To strengthen the point of a potential relationship between UBL3 and ciliary EVs, the authors should isolate sEVs and screen for UBL3. Alternatively, they could analyze a potential colocalization of UBL3 and sEV marker by using immunofluorescence microscopy.
Reviewer 3 Report
In this manuscript, Faber et al use pull downs and Mass Spectrometry to identify proteins interacting with PDE6D, a protein known to be important for ciliary localization of prenylated proteins and mutations in PDE6D are associated with the ciliopathy Joubert Syndrome. As starting material they used HEK293T cells transiently transfected with a PDE6D construct and mIMCD3 cells stably expressing the PDE6D construct. They find 42 unique proteins, significantly associated with PDE6D, of which 21 are prenylated. 14 proteins were identified in both HEK293T and mIMCD3 cells, of which 11 are prenylated. The identification of known interactors and ciliary proteins shows that their approach is valid.
Subsequently, the authors focus on three proteins for which ciliary localization had not been shown: FAM219A, NIM1K and UBL3. They confirm that these three proteins indeed interact with PDE6D in a prenylation-dependent manner. In addition, they show that NIM1K and UBL3 localize in cilia in a prenylation-dependent manner and that UBL3 also localizes in vesicle-like structures around the base of the cilium.
Next the authors identify proteins that interact with UBL3. These include proteins that regulate extracellular vesicles (EVs) and ciliogenesis. Finally, the authors show that UBL3 fused to GFP localizes to the outer segments of photoreceptors, at the distal part of the bulge region, extending into the distal axoneme and inside the connecting cilium. This localization depends on the prenylation-status of the protein.
Major comments:
- in the abstract, line 24, the authors state that NIM1K and UBL3 localize inside cilia in a PDE6D dependent manner. I don't think the authors show this in the manuscript. Please show these data or remove the claim.
- in the discussion, the authors extensively discuss the possibility that UBL3 and PDE6D might regulate extracellular vesicle formation or release (from cilia). I think this is indeed a good hypothesis, but as no functional experiments have been done to address this directly, I think this discussion should be significantly shortened. In addition, the insights of this manuscript would be significantly strengthened if such experiments would have been included, e.g. the analysis of knock out mice, as suggested by the authors, or knock down of UBL3 in any of the cells or tissues that they analyzed. Also overexpression of UBL3 could affect cilium length or extracellular vesicle formation or release, but the authors show now data analyzing this. Taken together, I'm not convinced that the new insights provided in this manuscript are sufficient for publication in Cells.
Minor comments:
- It seems Figure S1 is the same as Figure 1C. Please explain what the difference is.
- The authors refer to Figure 1C 'right', where I think they mean 'left'. Please correct.
- the authors use two mutant forms of the proteins that they study. First the prenylation defective mutant, which they refer to as MUT. Second they use mutants where the S is mutated to a K, which would prevent ciliary entry. When discussing the mutants and in the figures, it is unclear which mutants are referred to when the MUT nomenclature is used. Please use another name for these prenylation defective mutants, e.g. C>A.
- Please explain in the text briefly what amino acids are changed in the prenylation defective mutants.
- In lines 448 and 449 the S>K mutants are already introduced, but these results are only discussed in the next paragraph. Please remove these from lines 448 and 449, and only introduce them later on.
- In line 454 the others state 'Both mutant forms', suggesting they discuss the C>A and S>K mutants, but I think they actually mean only the C>A mutants. Please rephrase.
- line 497. Please explain what Tier II proteins are.
- line 523 states that UBL3 WT is mainly expressed.... This should be GFP::UBL3 WT mainly localizes to....
- Please indicate in Fig 5A where the different regions of the photoreceptors are.
- line 599 What do you mean with "possibly due to its recombinant overexpression"?
- line 607. I don't think the authors can claim that they "demonstrate that ... possibly also FAM219A is a novel ciliary protein". Their data is really not strong enough for this claim. It could very well be, but as they don't see it there, they cannot claim this. Have they tried immunofluorescence using antibodies directed against the endogenous protein?
Round 2
Reviewer 1 Report
The authors responded to most my comments satisfactorily. They have now mentioned quantification of the S>K mutants in text. However, the number of transfected/stable cells tested for eGFP tagged NIM1K and UBL3 WT/ C>A mutants have not been mentioned along with the numbers of cilia positive for these proteins. It will be good to add quantification of % positive cilia for these proteins before final publication.
Author Response
Reviewer 1 comment:
The authors responded to most my comments satisfactorily. They have now mentioned quantification of the S>K mutants in text. However, the number of transfected/stable cells tested for eGFP tagged NIM1K and UBL3 WT/ C>A mutants have not been mentioned along with the numbers of cilia positive for these proteins. It will be good to add quantification of % positive cilia for these proteins before final publication.
Response:
For most of the localizations we observed consistent localization either inside the cilium or at the basal bodies in all transfected ciliated cells, with the exception of eGFP-NIM1K S>K, as indicated before. For this reason, we mainly focused on the quantification of eGFP-NIM1K S>K. However, we do understand the comment of reviewer 1 to also indicate the numbers of the other mutants. For the NIM1K and UBL3 S>K mutants we analyzed 25 cilia of transfected cells, as indicated before. The localization of both the WTs and C>A mutants were fully consistent. Therefore, we decided that analysis of at least 15 transfected ciliated cells is sufficient to draw conclusions. Immunostainings were performed in duplicate and repeated at least twice.
In the updated manuscript, we added the following paragraph to the materials and methods; section 2.8:
For protein localization, we analyzed at least 15 transfected ciliated cells for eGFP-FAM219A WT/C>A, eGFP-NIM1K WT/C>A, and UBL3 WT/C>A. For localization of eGFP-NIM1K S>K and eGFP-UBL3 S>K, a total of 25 transfected ciliated cells were analyzed. Immunostainings were performed in duplicate in at least two biological replicates.
Reviewer 2 Report
The authors have done a good job in addressing my comments.
Author Response
We thank the reviewer for the positive comment.